# Epothilones as Natural Compounds for Novel Anticancer Drugs Development

**DOI:** 10.3390/ijms24076063

**Published:** 2023-03-23

**Authors:** Cecilia Villegas, Iván González-Chavarría, Viviana Burgos, Héctor Iturra-Beiza, Henning Ulrich, Cristian Paz

**Affiliations:** 1Laboratory of Natural Products & Drug Discovery, Center CEBIM, Department of Basic Sciences, Universidad de La Frontera, Temuco 4811230, Chile; 2Departamento de Fisiopatología, Facultad de Ciencias Biológicas, Universidad de Concepción, Concepción 4030000, Chile; 3Departamento de Ciencias Biológicas y Químicas, Facultad de Recursos Naturales, Universidad Católica de Temuco, Temuco 4800000, Chile; 4Departamento de Ciencias Básicas, Facultad de Ciencias, Universidad Santo Tomas, Temuco 4780000, Chile; 5Department of Biochemistry, Instituto de Química, Universidade de São Paulo, Av. Prof. Lineu Prestes 748, São Paulo 05508-000, Brazil

**Keywords:** epothilones, epothilone derivates, cytotoxicity, anticancer agents, refractory cancer, taxanes, clinical trials

## Abstract

Epothilone is a natural 16-membered macrolide cytotoxic compound produced by the metabolism of the cellulose-degrading myxobacterium *Sorangium cellulosum*. This review summarizes results in the study of epothilones against cancer with preclinical results and clinical studies from 2010–2022. Epothilone have mechanisms of action similar to paclitaxel by inducing tubulin polymerization and apoptosis with low susceptibility to tumor resistance mechanisms. It is active against refractory tumors, being superior to paclitaxel in many respects. Since the discovery of epothilones, several derivatives have been synthesized, and most of them have failed in Phases II and III in clinical trials; however, ixabepilone and utidelone are currently used in clinical practice. There is robust evidence that triple-negative breast cancer (TNBC) treatment improves using ixabepilone plus capecitabine or utidelone in combination with capecitabine. In recent years innovative synthetic strategies resulted in the synthesis of new epothilone derivatives with improved activity against refractory tumors with better activities when compared to ixabepilone or taxol. These compounds together with specific delivery mechanisms could be developed in anti-cancer drugs.

## 1. Introduction

Reichenbach, Höfle, and co-workers from the German National Center for Biotechnology Research (GBF) [1,2] first extracted epothilones (Epo) in 1987. Epo are natural products produced by the cellulose-degrading myxobacterium *Sorangium cellulosum* SoCe90, with potent activity against different types of cancer and action mechanisms similar to those of taxanes. Structurally, epothilones and taxanes are unrelated as shown Figure 1. However, both compounds promote microtubule assembly and stability, sharing a common β-tubulin-binding site. The Epo structure consists of a 16-membered lactone macrocycle, which includes an epoxide between C12 and C13, a ketone at C5, and a thiazole moiety in a side chain at C15. Epothilone A (EpoA) and Epothilone B (EpoB) were the first to be discovered. They have similar structures, with the exception that EpoB has an additional methyl group at C12 [3]. Before the discovery of EpoA and EpoB, approximately 37 other natural epothilones have been identified, of which only C, D, E, and F are structural analogs of epothilones A and B, with discrete differences and activities. While epothilones G and H possess the typical 16-membered ring, they also contain an oxazole moiety instead of a thiazole in the side chain [4].

The discovery of mechanisms of Epo action provided an alternative for the development of novel antimitotic drugs different from taxanes, which are expensive starting materials with limited chemical modification possibilities, poor water solubility, high toxicity, and fast resistance development by tumors.

Paclitaxel (Taxol^®^) is a natural member of the taxane family and the most widely used in the treatment of cancer, is a tetracyclic diterpenoid compound (Figure 1), originally isolated in the 1960s from the bark of Pacific yew trees (*Taxus brevifolia*), but currently produced by artificial cultivation of *Taxus* plants, microbial fermentation, and chemical hemisynthesis, as well as genetic engineering [5]. Because of its high efficacy and broad spectrum against cancer cells, this drug has played a crucial role in the treatment of ovarian cancer, breast cancer, uterine cancer, non-small cell lung cancer, and more recently, metastatic pancreatic cancer and Kaposi’s sarcoma [5,6]. Its use is limited by the development of resistance mechanisms and the development of adverse effects related more directly to the use of excipients than to the compound itself, as Cremophor^®^ EL (CrEL) [6]. This vehicle promotes side effects, including hypersensitivity, nephrotoxicity, and neurotoxicity. Consequently, all patients receiving Taxol^®^ should be pretreated with corticosteroids, H2 antagonists, and antihistamines to prevent hypersensitivity reactions, which can even be fatal [6]. The low solubility of Taxol (0.3–0.5 μg/mL) forces the use of formulations for their administration [7]. 

Epothilones, unlike paclitaxel, the usual taxane drug, do not need to be dissolved by adjuvants that can be toxic in some patients. They have good plasma solubility, are also active against resistant tumor cells, and their structure is more amenable to chemical manipulation, allowing the production of epothilone analogs, with improved physicochemical properties [8], which have been the focus of developing new therapeutic agents against cancer multidrug resistance.

Currently, more than 400 epothilone derivatives have been reported [9] and classified into three major groups: first-generation epothilones, which include only natural products; second-generation epothilones, which are semi-synthetic derivatives; and third-generation epothilones, or fully synthetic derivatives [10]. Epothilone B and D have been the most promissory compounds for the development of anticancer drug derivatives. Six compounds have been clinically evaluated in humans: Epothilone B, also called patupilone or EPO 906, ixabepilone (BMS-247550), BMS-310705, sagopilone (ZK-EPO), epothilone D (utidelone, UTD1, KOS-862), and fludelone (KOS-1584), a second generation epothilone (Figure 2). 

In this sense, the term “Ephotilone” in the clinical trials database to date, yields a total of 114 results, of which the common condition is some type of cancer caused by solid tumors, Ixabepilone being the only one that has passed Phase 3, obtaining U.S. Food and Drug Administration approval as monotherapy and in combination with capecitabine for the treatment of metastatic or locally advanced refractory breast cancer [11]. EpoB or Patupilone is the natural epothilone with activity against taxane-resistant cells [12], and most of the knowledge about the anticancer mechanism of action of epothilones is based on this compound. However, despite the efficacy against a broad number of cancer cell including resistance lines [13] and in vitro trials [14], its clinical use has been limited because of the development of adverse effects, failing in Phase III trials. Recently, Utidelone (UTD1), a genetically modified epothilone analogue, has demonstrated excellent efficacy in Phase II and III trials in terms of disease-free survival (DFS) in patients with metastatic breast cancer, with a lower incidence of induced neuropathy [15].

The production of epothilones have improved over time, and now the biotechnology mediated by TALE-TF and CRISPR/dcas9 Systems [16] or the use of other microorganisms with putative genes that have a high similarity to the *Sorangium cellulosum* cluster, such as *Aspergillus fumigatus* EFBL [17], has increased the production of these compounds. Moreover, the use of organic synthesis and the design of non-natural analogs or the production of analogs by manipulation of the epothilone biosynthetic gene cluster achieve attractive methods for getting new derivatives or starting materials for synthesis [10,18,19].

This review summarizes preclinical findings and clinical studies of epothilones in cancer patients, as well as advances in the design and synthesis of new epothilone derivatives. Data were identified by searching in MEDLINE/PubMed, Google Scholar, and American Society of Clinical Oncology publications using the terms “epothilone, cancer, and Clinical trial” between 2010 and 2022.

## 2. Epothilone Synthesis

The first total synthesis of epothilone was reported in 1996/1997 by the research groups of Danishefsky, Nicolaou, and Schinzer [20,21,22]; after that, many researchers focused in the synthesis of epothilones by different approaches as the use of immobilized reagent techniques [23,24], by Stille coupling approach [21,25], by Nerol/macroaldolization approach [26], by multifunctional asymmetric catalysis [27], by antibody catalysts [28], or by ring-closing metathesis [29]. For example, Cheng and Huang [10] obtained epothilone in 23 steps, whereas Wang, et al. [30] synthesized (−)-epothilone B over 11 steps with a total yield of 8% ca. The epothilone synthesis pathway uses expensive reagents and some steps with a moderate yield making yet a large epothilone production unviable by chemical synthesis. 

The original fermentation procedure used for epothilone production by Reichenbach, Höfle, and co-workers shows some inconvenience [2]. *S. cellulosum* has long fermentation cycles, around 1 month, an irregular epothilone production, and is difficult to genetically manipulate. 

The epothilone biosynthetic pathway involved the mixed via polyketides synthase type I (PKS), as well as the non-ribosomal peptide synthetase (NRPS) complex [31]. PKS modules (EpoA, EpoC, EpoD, EpoE, and EpoF) produce the epothilone backbone, whereas the NRPS module (EpoB) catalyzes the thiazole ring synthesis of epothilone from cysteine [32]. Finally, a P450-epoxidase (encoded by epoK) is responsible for the C12-13 epoxidation of epothilone C and D giving epothilone A and B, respectively. Epothilones are produced not only by *S. cellulosum*. El-Sayed, et al. [17] found that endophytic fungi from medicinal plants with conservative epoA PKS-NRP domains also produce epothilones. For example, *A. fumigatus* isolated from *Catharanthus roseus*, produces epothilones with a yield of 21.5 μg/g biomass, and the fungus *Alternaria alternata* isolated from *Coriandrum sativum* produces 11.6 μg/g biomass of epothilones [17]. 

The genetic modification of *Sorangium cellulosum* as well as the expression of the epothilone gene cluster in heterologous microorganisms has increased the production of epothilone by fermentation arising to levels that make viable a large-scale production, Table 1.

## 3. Epothilone Induce Stabilized Microtubule Assembly

Epothilones displays cytotoxic effects against a large number of cancer cells at nanomolar concentrations with a similar effect to paclitaxel, causing the arrest of mitosis by promoting microtubule polymerization in the absence of GTP or microtubule-associated proteins under conditions that promote depolymerization such as dilution, low temperatures, or Ca^2+^ [39]. The microtubule stabilized by epothilone has a reduced number of protofilaments and a smaller diameter than normal [39,40], and it is dysfunctional and aberrantly present during the M phase, leading to cell cycle arrest at metaphase/anaphase transition and cell death [41].

The interaction of epothilones for binding to β-tubulin, occurs in a close overlapping binding site to taxol, which was previously studied by fluorescence-based displacement assay. Natural epothilones displace ^3^H-taxol from tubulin with a binding constant of 2.93 × 10^7^ M^−1^ and 6.08 × 10^8^ M^−1^ for Epo A and B, respectively, and Ki values of 1.4 and 0.7 mM [42,43], suggesting that paclitaxel and epothilones share a common or similar binding place. Epothilone B is a more efficient polymerizing agent of tubulin even than Epo A and paclitaxel, which can displace the labeled ^3^H-paclitaxel form the ^3^H-paclitaxel/microtubule complex [44]. Further studies by Prota et al., 2013 on a crystallography complex between αβ-tubulin, the statin-like protein RB3, tubulin–tyrosine ligase, and epothilone A (2.3 Å resolution) showed that EpoA binding to tubulin in the taxane pocket of β-tubulin, formed by hydrophobic residues of helix H7, β S7 strand, the M-loop, and the S9–S10 strands of β-tubulin, interfering with the correct microtubule architecture of protofilaments and the tubulin assembly equilibrium [45]. Similar interactions have been proposed for Ixabepilone, EpoB, and Epotilone D, establishing that the interaction occurs via H-bond networks with the residues Thr276 and Gln281 in the M-loop and the hydrophobic surface–surface interactions with Leu217, Ile212, Leu275, Leu30, Phe272, and Leu371 in the binding pocket [46]. In addition to the crystallography results, a biological study using an epothilone photoprobe identified the peptides TARGSQQY and TSRGSQQY (from amino acids 274 to 281) as the specific region of interaction between β-tubulin and epothilones, which is consistent with that previously reported by X-ray crystallography [47].

The understanding of the interactions between epothilones and microtubules allows the creation of new derivatives with increased potency; therefore, a subdivision of the molecule into four regions (A–D) based on structure–activity relationship (SAR) was proposed [25] (see Figure 3). In this regard, modifications in epoxide (C12-13) or near in B region are known to significantly affect microtubule-stabilizing activity [48]; for example, EpoB possessing an additional methyl group at C12 exhibits twice as potent activity as EpoA in inducing tubulin polymerization in vitro [42]. It has been recognized as essential for the activity of epothilones: the C7 hydroxyl in the A region of the molecule, the aromatic substituents at C-15 together with their chirality (Region C), and the C1 carbonyl in D region [9,43]. It is known that the C12-13 epoxide is not essential for binding to microtubules because deoxyepothylone B (Epothilone D, UTD1 or KOS-862) lacking the epoxide, is more active in microtubule stabilizing in vitro when compared to epothilone A or B. Furthermore, derivatives containing nitrogen at C12 (azathilones) show higher pharmacological activity in vitro [10,49].

## 4. Epothilones Induce Apoptosis in Multidrug-Resistant Cancer Cells

Epothilones induce cell death by apoptosis in a mechanism that occurs after 48 h of drug exposure, and because of the cell cycle arrest at the metaphase/anaphase transition, these findings are well established regarding the effect of EpoB and its derivatives [50]. However, molecular mechanisms explain that the apoptosis induction by epothilones can occur by both extrinsic and intrinsic pathways, depending on the type of cell line and the functional status of the *TP53* gene [51,52]; for example, EpoB increase the expression of p53 in A549 cells (lung carcinoma) that possess wild-type TP53 [52]. The activation of p53 is an early event of the intrinsic apoptotic pathway that leads to the activation of different proteins and apoptogenic factors, which would explain the report by Lee, et al. [53] who observed in the human colon cancer cell line SW620 (p53^+^) that EpoB can trigger the expression of the active form of caspase-3 together with the proapoptotic proteins Bax, p53 via an NF-κB-dependent pathway and at the same time, reduce the expression of the antiapoptotic protein Bcl2 [53]. Moreover, poly (ADP-ribose) polymerase 1 (PARP1) excision, which together with caspase activation constitutes a hallmark of both intrinsic and extrinsic apoptosis, was detected in p53-deficient FaDu cells (squamous cell carcinoma) treated with EpoB [52] and in RKO cells (p53^+^) treated with UTD1 [54]. In this case, the induction of apoptosis occurred through the activation of caspase-3 and PARP as an increased ROS level consequence, decreased mitochondrial membrane potential, and activation of c-Jun N-terminal kinase (JNK), suggesting that the ROS/JNK pathway is involved in this process [54]. 

Several studies indicate that mitochondrial events, such as decreased membrane potential caused by ROS, play a key role in epothilone-induced apoptosis, which is more evidenced by EpoB compared to paclitaxel [55]. However, it is also suggested that the mitochondrial imbalance could originate from the increased ROS levels, as well as from intracytoplasmic Ca^2+^ concentrations ([Ca^2+^]). In this sense Rogalska, et al. [56], evaluated the possible role of calcium in the mitochondrial cascade events in human SKOV-3 after treatment with EpoB; finding that 24 h after treatment with IC50 of EpoB, a significant increase in [Ca^2+^] (21%) occurs, followed by the release of Cytochrome C which is a critical proapoptotic event. These increases in the intracellular calcium concentration are not induced by paclitaxel. Nevertheless, ROS has been reported as a central element in the early induction of mitochondrial membrane potential alterations and thus an important trigger of Cytochrome C release, in this sense, human neuroblastoma cells treated with EpoB observed a reduction of mitochondria-confined cytochrome C by 66% after 6 h of treatment, with a maximum reduction in 83% at 24 h [57]. Poly (ADP-ribose) polymerase (PARP) lacking functional p53 indicates activation of apoptosis via the extrinsic pathway, since activation of caspase 3 upstream of PARP excision can be triggered by caspase 8, responsible for death receptor signaling cascade beginning. Rogalska and Marczak [58] observed in OV-9 (ovarian cancer cells) that EpoB triggers TRAIL/caspase 8-dependent apoptosis, an apoptosis pathway that simultaneously causes mitochondrial alteration. TRAIL (TNF-related apoptosis-inducing ligand) induces apoptosis preferentially in malignant cells, whereas it does not affect normal tissue, and therefore may play a selective role in regulating susceptibility to cell apoptosis [59]. Related to this, Cisplatin and paclitaxel-resistant ovarian cancer cells treated with ixabepilone exhibit increased expression of death receptors DR4 and DR5, along with increased APO-2L/TRAIL-induced caspase 8, indicating activation via the extrinsic apoptosis pathway [60]. Table 2 summarizes the epothilone apoptosis mechanism. 

Resistance to antineoplastic agents can be caused by numerous cellular mechanisms, such as activation of drug metabolizing enzymes and DNA repair mechanisms, blockade of apoptotic signaling, and augmented activity of drug efflux pumps, resulting in multidrug resistance (MDR) [62]. MDR is the term that describes the ability of drug-resistant tumors to show simultaneous resistance to several structurally and functionally unrelated chemotherapeutic agents, often mediated by the overexpression of drug efflux pumps, such as P-glycoprotein (Pgp), MDR protein (MRP-1), and breast cancer resistant protein (BCRP) [63,64]. Paclitaxel and docetaxel are substrates of P-gp; therefore, in tumor cells overexpressing P-gp, the intracellular concentration of these drugs decreases rapidly, which is the most common mechanism of resistance [65]. In this sense, overexpression of MDR-1 is frequent in some human cancers and has been associated with chemoresistance to taxanes [66], a negative prognostic in many cancers, as acute myeloid leukemia, breast cancer, osteosarcoma, bladder tumor, ovarian cancer, and central nervous system and many other tumors [67].

Epothilones, unlike taxane derivatives, are poor substrates for P-gp, and for that reason they are active in numerous taxane-resistant cancer cell lines. Moreover, MDR protein expression is not altered in epothilone-resistant models in vitro. In Phase III clinical trials, ixabepilone and UTD1 have shown efficacy in the treatments of patients with metastatic and/or refractory breast cancer [68]. An overview of epothilone activity is shown in Figure 4. 

## 5. Epothilones with Clinical Significance in Cancer

EpoB derivatives such as Ixabepilone and UTD1 are currently used in the clinic for the treatment of advanced breast cancer, their efficacy has been demonstrated in Phase II and III clinical trials. 

### 5.1. Epothilone B

EPO906 or patupilone is a synthesized version of natural Epothilone B (EpoB) developed by Novartis, with the same structure and, therefore, the same mechanism of action [69]. EPO906 has been part of many preclinical [13,14,70] and clinical studies [71,72,73]. Due to its conserved activity against taxane-resistant cells, it could be an alternative for patients who relapse after chemotherapy based on paclitaxel and platinum derivatives and present high levels of class III beta-tubulin (TUBB3). Clinical trials results recognize the potential of patupilone to improve progression-free survival and pain response in castration-resistant prostate cancer and gynecologic cancer, whose relapse events are accompanied by resistance to docetaxel and cisplatin, respectively. Other tumor pathologies, such as colorectal cancer (CRC) [74] and brain metastases [75], have been evaluated but without significant evidence of efficacy.

In castration-resistant prostate cancer (CRPC), patupilone showed variable results either as first-line therapy (compared to standard docetaxel therapy) or in docetaxel-refractory patients. For instance, a Phase II clinical study was conducted with 45 patients (64% pretreated with taxanes) who received patupilone (2.5 mg/m^2^) via 5-min bolus i.v. infusion, once per week for 3 weeks, followed by one week of rest (4-week cycle). Decreased patient prostate-specific antigen (PSA) levels by ≥50% was confirmed in only six patients (13%), and the mean time to progression was 1.6 months, which was considered as insignificant [76]. In a second study with 83 participants injected with 10 mg/m^2^ patupilone by i.v. every 3 weeks, a PSA level decrease of ≥50% was observed in 47% of the patients. Pain responses were observed in 59% of evaluable patients. The median time to PSA level progression was 6.1 months, and the median overall survival (OS) was 11.3 months (95% CI: 9.8 to 15.4), demonstrating antitumor activity and contribution to symptomatic improvement in patients previously treated with docetaxel [77].

The clinical efficacy of patupilone was also evaluated in castration-resistant prostate cancer (mCPRC) patients who had never received chemotherapy, these patients with mCPRC have a poor prognosis, and first-line therapy is mainly based on docetaxel-based regimens which makes them prone to develop resistance quickly [78]. The treatment based on patupilone + prednisone revealed a better response in view of PSA levels, compared to the classical treatment of docetaxel + prednisone [79] in patients who received intravenous (I.V.) patupilone (10 mg/m^2^ or docetaxel 75 mg/m^2^ every 3 weeks. 

In ovarian cancer, patupilone has been tested clinically with Phase I and II studies. Novartis Pharmaceuticals evaluated patupilone in a randomized open-label Phase III study (NCT00262990) in 829 patients with interracial differences from 11 different countries. This analysis compared the efficacy of patupilone versus liposomal doxorubicin in patients with epithelial ovarian cancer. The results showed a modest efficacy profile in patients treated with patupilone, but the treatment did not achieve a significant improvement in overall survival compared to the active control [80]. The most frequent toxicity in the patupilone arm was diarrhea (25.6% grade 3 and 4) and mild peripheral neuropathy (6.2% grade 3 and 4), consistent with Phase I and II trials of monotherapy where diarrhea grade 3 was the most common adverse effect present in 8–29% of patients with the highest incidence in patients with colorectal and prostate cancer [81], and is the dose-limiting event in therapeutic schemes. 

The safety and toxicity profile of epothilones has been the main problem for bringing these drugs to market, ixabepilone, KOS-862, and ZK-EPO (sagopilone) exhibit a high incidence of myelosuppression, alopecia, severe peripheral neuropathy, and hypersensitivity reactions [82,83]. Despite patupilone (EpoB) showing less toxicity than its derivatives, receiving a good opinion from the European Medicines Agency (EMEA) for the treatment of cancers of the female reproductive system, the effectivity on patients was not significant compared to the control, failing in the Phase III trial [80]. 

Patupilone, unlike taxanes, has shown the ability to cross the blood–brain barrier (BBB) and exert antitumor effects in brain tumors. In vitro patupilone evaluated in animal models [84] was able to reduce the proliferative activity of medulloblastoma cell lines at picomolar concentrations (50–200 pM), and it produced an anti clonogenic effect in combination with ionizing radiation (2 or 5 Gy) [85]. 

In humans, patupilone accumulation in glioblastoma (GBM) tumor tissue is 30 times higher compared to plasma values at 20 min. In this sense, administration of patupilone before and after surgery in recurrent GBM is safe, improving long-term PFS in patients, and could be an alternative for the treatment of central nervous system (CNS) cancers [86]. However, larger clinical trials are needed to ensure the safety and efficacy of this drug. 

Patupilone promotes neural regeneration after an injury, inducing concerted polymerization of microtubules at the axon tip, which drives axon growth by inhibiting the migration scar-forming fibroblasts and reactivating neuronal polarization [87].

### 5.2. Ixabepilone

Ixabepilone, also called aza-epothilone B or BMS 247550 is a semi-synthetic second-generation analogue of the natural product EpoB, which changed the macrolide lactone ring, with nitrogen to give the corresponding macrolactam [88]. The improvements of the semi-synthetic compound include a higher antitumor activity than epothilones A and B against a broad spectrum of human tumors. It was developed by Bristol-Myers Squibb (BMS, New York, NY, USA), marketed in the USA under the name Ixempra^®^, and was listed by the FDA in the USA in 2007 for the treatment of metastatic or advanced breast cancer, either as a single agent or together with capecitabine for the treatment of patients with metastatic or locally advanced breast cancers thatshow resistance against anthracycline and taxane treatment [89]. Nevertheless, the Committee for Medicinal Products for Human Use (CHMP) recommended further research evaluating the risk–benefit ratio in the use of the drug due to the high incidence and neurotoxicity with its use.

Early preclinical trials demonstrated that ixabepilone has been shown to induce the activation of selective apoptotic pathways [90]. The compound is effective against multiple cancers, including those tumors resistant to common chemotherapeutic agents, such as against paclitaxel-resistant lines HCT116/VM46 (colorectal cancer), Pat-21 (ovarian carcinoma), Pat-7 (breast), and A2780 Tax (ovarian carcinoma), which express tubulin mutation, as well as to sensitive lines Pat-26 (human pancreatic carcinoma) and M5076 (murine fibrosarcoma) displaying a cytotoxic effect around 2.9 nM (IC_50_) [91] (see Table 3). Ixabepilone also shows low susceptibility to multiple resistance mechanisms because it is a poor substrate of P-gp, which is overexpressed in malignant neoplasms of solid tumors, as kidney, colon, liver, ovary, breast, and sarcomas [63]. In this sense, being active against pediatric solid tumor cell lines of osteosarcoma (HOS), Ewing’s sarcoma (LD-EWS), and rhabdomyosarcoma (RD), evidencing a similar potency to paclitaxel, vincristine and vinorelbine, the standard tubulin-binding anticancer drugs [92]. 

In Phase II studies, ixabepilone was effective against hormone-refractory prostate cancer (HRPC) [93]; non-small cell lung cancer (NSCLC) [94], including NSCLC tumors that have failed in the first-line platinum-based chemotherapy [95]; and other resistant cancers, such as renal [96] and pancreatic carcinomas [97]. However minimal effects have been observed in gynecological cancer [98,99].

The treatment with ixabepilone plus capecitabine demonstrates superior efficacy in terms of PFS to capecitabine alone, in patients with anthracycline- or taxane-resistant metastatic breast cancer [100,101], as well as in patients with triple-negative breast cancer (TNBC) where the combination of ixabepilone with capecitabine approximately doubles the median PFS [102], comparable to those observed in non-triple-negative tumor patients [103]. This is particularly advantageous in this patient population because TNBC accounts for 15–20% of all breast cancer and is associated with shorter survival after metastasis development. At present, targeted therapies exist for this type of cancer, and therefore, chemotherapy remains the primary treatment [104]. Controversial results were seen in the Phase II clinical study, where the efficacy of ixabepilone alone or together with ixabepilone plus cetuximab as first-line treatment in patients with advanced/metastatic TNBC showed no significant differences [105]. However, the reported study TITAN, evaluating ixabepilone substitution for paclitaxel after doxorubicin/cyclophosphamide (AC) in adjuvant treatment of early-stage TNBC, revealed similar DFS and OS in patients with operable TNBC compared to AC/paclitaxel treatment, but with less marked adverse effects in contrast to paclitaxel, which could mean an alternative for second-line treatment in these types of patients [106].

Utidelone, UTD1, KOS-862, or Epothilone D is an epothilone derivative generated by genetic engineering of the epothilone gene cluster, increasing the concentration of UTD1 by fermentation in *S. cellulosum.* It was developed and manufactured by Biostar Technologies, Ltd., Beijing, China. UTD1 has revealed strong in vitro and in vivo activity against paclitaxel-sensitive tumors, such as multidrug-resistant human colon, leukemia, and breast tumors [107]. 

UTD1 is currently an alternative for the treatment of metastatic breast cancer (MBC), especially in breast cancer previously treated with anthracyclines and taxanes [108], as well as for patients with HER2-positive breast cancer [109]. Phase II clinical studies developed in Asia evaluated its efficacy, showing positive results, promising tolerability, and advantageous safety profiles in patients who completed a median of six cycles of therapy alone or in combination with capecitabine. In this regard, combination therapy has shown better results than therapy alone when evaluating the objective response rate (ORR) and PFS. The combination therapy yielded an ORR of 42.4% and a median PFS of 7.9 months, whereas the monotherapy study resulted in an ORR of 28.57% and a median PFS of 5.4 months [108].

Results of a Phase III randomized controlled trial evaluating (OS) in heavily pretreated MBC, refractory to anthracyclines and taxanes, supported the use of UTD1 plus capecitabine as a novel therapeutic regimen for these patients. The evaluation of 405 patients who received UTD1 (30 mg/m^2^ IV daily, days 1–5, for 90 min) plus capecitabine (1000 mg/m^2^ orally b.i.d., days 1–14) or capecitabine alone (1250 mg/m^2^ orally b.i.d., days 1–14) every 21 days, ratified the improvement in OS in the combination group (19.8 months) compared to the monotherapy group (16.0 months), demonstrating that combination therapy with UTD1 remained superior to capecitabine monotherapy [110]. This combination therapy was included in the Chinese Society of Clinical Oncology (CSCO) Breast Cancer Guidelines 2022 recommendations for salvage treatment of triple-negative breast cancer as a level II recommendation, which includes protocols with a relatively high level of evidence, but where a slightly lower expert consensus is used [111].

Similar results were obtained for HER2-positive (human epidermal growth factor receptor 2) breast cancer therapies. The Phase 2 study (NCT04681287) evaluated UTD1 in patients who have been pretreated with trastuzumab and tyrosine kinase inhibitors. Participants received intravenous camrelizumab (200 mg once every 3 weeks), inetetamab (loading dose of 8 mg/kg and then 6 mg/kg, day 1), and UTD1 (30 mg/m^2^, days 1–5) until the disease progressed or intolerable toxicity occurred. All three drugs showed promising efficacy and an acceptable safety profile, representing a new option for this type of patients [109].

UTD1 has been proposed for use in lung cancer in which chemotherapy is the gold standard treatment in most patients. The Phase II clinical investigation with patients enrolled between 2019 and 2021 (NCT03693547) reported that UTD1 is safe for advanced NSCLC refractory to second-line treatment and could be effective; however, more studies are needed in this specific type of cancer [112].

The activity of UTD1 for the treatment of colorectal cancer (CRC) currently being studied. In this regard, UTD1 has exhibited broad antitumor activity in RKO and HCT116 cells, as reported by [54]. UTD1 inhibited CRC cell proliferation, in vitro, with an IC50 of 0.38 µg/mL and 0.77 µg/mL against RKO and HCT116, respectively. These results were also reproducible in RKO xenografts in nude mice, suggesting that UTD1 could be an effective agent in the treatment of CRC in humans. The mechanism of action is similar to that described for other epothilones: induction of microtubule cluster and aster formation, inducing cell cycle arrest in the G2/M phase, and subsequent apoptosis. UTD1 exhibited stronger apoptosis induction effects than paclitaxel and 5-FU, especially in HCT15 cells with ABCB1 overexpression. In CRC cells, UTD1 increased ROS production along with activation of c-Jun N-terminal kinase (JNK), suggesting a mechanism through the ROS/JNK pathway [54].

## 6. Toxicity and Safety Profile of Epothilones

Safety and toxicity profiles are different according to the type of epothilone. For instance, ixabepilone, KOS-862 (Epothilone D), and ZK-EPO (sagopilone) show high incidences of myelosuppression, alopecia, peripheral neuropathy, and hypersensitivity reactions [82,83], whereas patupilone has a milder toxicity profile characterized by diarrhea and fatigue [82,83]. These differences have been attributed to the tissue distribution which is given by the structural characteristics of the molecule as well as the formulation [113]. Ixabepilone is the most widely used epothilone in clinical practice, and therefore, the most reported. The regimens with 40 mg/m^2^ of ixabepilone can result in adverse effects that mark the limiting dose. The most common effects in almost all proposed regimens in Phase I trials of ixabepilone in monotherapy were peripheral neuropathy and asthenia/fatigue. In Phase II studies, the highest reported adverse effects (3/4) are leukopenia (36% grade 3 and 13% grade 4), sensory peripheral neuropathy (SPN, 2–20% grade 3 and 0–1% grade 4), and fatigue/asthenia (6–27% grade 3 and 0–1% grade 4) reported mainly in patients with pretreated metastatic breast cancer [114].

The peripheral nervous system may be vulnerable to the toxic action of various drugs because it is not as effectively protected as the central nervous system against exogenous noxious agents. Peripheral neurotoxicity of antineoplastic agents is dose-limiting side effects, which is given by the ability of the drugs to affect nerve fibers or neuronal bodies such as dorsal root ganglia of primary sensory neurons [115]. Ixabepilone-induced peripheral neuropathy is generally cumulative, reversible, and can be controlled by dose reduction [116,117]. It has been reported that up to 88% of patients exposed to ixabepilone treatment may manifest SPN of any grade, whereas the incidence of grade 3/4 sensory neurotoxicity may range from 6% to 24%; Motor neuropathy of any grade may be present in up to 16% of patients, but grade 3/4 only occurs occasionally in 0–5% [83].

It is important to mention that the reports of ixabepilone-induced SPN are data that have been obtained mostly from patients with pretreated and taxane-resistant BCM [118]. In this regard, it is documented that patients suffering from neuropathy caused by treatment prior to treatment with another chemotherapy regimen are vulnerable to develop more severe SPN [119], which is a predisposing factor in the case of patients treated with ixabepilone. This may be the reason for the high percentage of patients manifesting neurotoxicity following the administration of ixabepilone (40 mg/m^2^) every three weeks [83], which is significantly higher compared to the development of SPN grade 3 in non-pretreated patients. Yardley, et al. [120] reported, in a Phase II trial (N = 168), that for HER2-negative breast cancer patients treated with ixabepilone and cyclophosphamide as neoadjuvant therapy, only 8% of them developed SPN grade 3 [121]. Ixabepilone is tolerated at the approved dose (40 mg/m^2^ every 3 weeks) as SPN is a reversible and controllable event [116]. However, in more severe cases, the dose reduction of ixabepilone to a weekly schedule is effective in terms of OS and PFS and shows reasonable tolerability, improving the risk/benefit profile [122], even in patients with severe neuropathy leading to treatment discontinuation, experienced a better response rate (79%), longer PFS (11.3 months), and better OS (36.6 months).

## 7. New Epothilone Derivatives with Increased Cytotoxic Activity

At the moment, ixabepilone has been approved by the FDA for clinical use, whereas UTD1 is used in China. These two compounds are only a few examples of the rational or combinatorial synthesis of new epothilone derivatives. Nicolaou, et al. [123] used the structure of EpoB as a starting point for the synthesis of new compounds with better cytotoxic potencies, including different motifs such as fluorine, aziridine moiety, heterocyclic side chain, replacement of the epoxide for a difluorocyclopropyl moiety, or ixabepilone analogs, producing 54 compounds, which were evaluated against cancer cell lines. The most potent compounds are presented in Figure 5. 

Table 4 summarizes the activity of the most potent synthetic epothilones against MES SA DXE: multidrug-resistant uterine sarcoma; MES SA DX: human uterine sarcoma with marked multidrug resistance; HEK 293T: human embryonic kidney cell line. SKBR3: human breast cancer cell line; SKOV3: human ovarian cancer cell line and HeLa: human cervical carcinoma cell line.

Compound 9, which replaces the original epoxide in EpoB for an aziridine moiety, has a remarkable activity, several folds more portent against all cancer cell lines than monomethyl auristatin E. Previously, Nicolaou, et al. [121] published the synthesis of 12, 13-aziridinyl epothilones as potent antitumor agents, synthesizing 81 compounds and testing them against the cancer cell lines MCF-7; OVCAR-8; NCI/ADR-RES; MDA-MB-435; SNB-75; MES SA; MES SA DX; and HEK 293. The structure of the best compounds is given in Figure 6, and their activities are summarized in Table 5. 

These findings open the possibility for more potent anticancer epothilone derivatives which must bestudied further to assess the pharmacological properties in clinical studies.

## 8. Conclusions

The taxol-based chemotherapy is inefficient in the treatment of some tumors such as prostate, lung, ovarian, and breast cancer due to MDR development, pointing to the urgent need for the discovery of new anticancer drugs. Epothilones induce microtubule stabilizing effects with a similar mechanism to taxanes with the advantage that maintaining its cytotoxic activity in resistant cell lines that overexpress MDR mechanisms. Ixabepilone and UTD1 are currently in clinical use in combination with classical antitumoral drugs, improving cancer therapy and improving prognosis in patients. Genetic engineering has improved the production of epothilones by fermentation procedures increasing the yields substantially and making suitable the biotechnological production rather than by total synthesis, facilitating the therapeutic application of epothilones. The simple structure of epothilones has allowed the synthesis of hundreds of derivatives, some of them with many fold higher activities than EpoB, ixabepilone, or even taxol. In view of that, some of these compounds are promising for cancer therapy. However, more clinical assays need to be performed ensuring their safety and activity. 

## Figures and Tables

**Figure 1 ijms-24-06063-f001:**
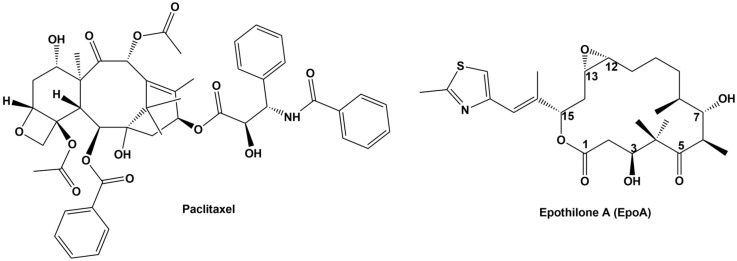
Chemical Structure of the natural Epothilone A and the taxane Paclitaxel.

**Figure 2 ijms-24-06063-f002:**
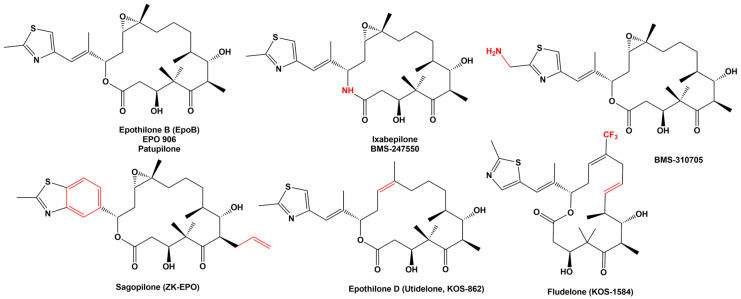
Structure of natural and synthetic epothilone with anticancer activity clinically evaluated. In red are remarked the main differences with EpoB.

**Figure 3 ijms-24-06063-f003:**
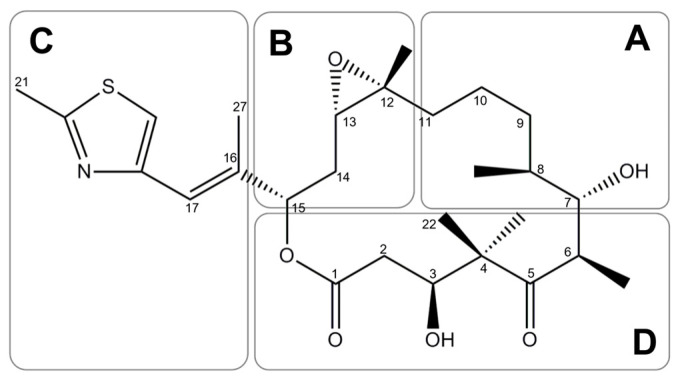
Subdivision of the EpoB structure into four regions (**A**–**D**) based on structure activity relationship [9,25].

**Figure 4 ijms-24-06063-f004:**
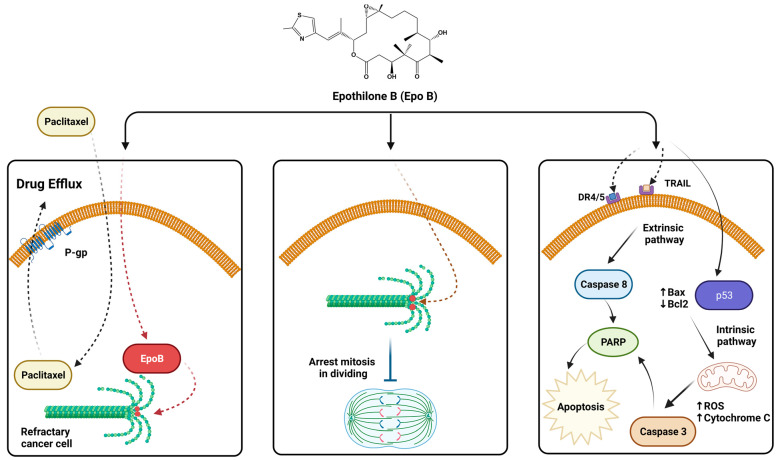
Mechanisms of action of epothilones. (**left**) Evasion of MDR-associated resistance. (**centre**) Stabilization of microtubules and inhibition of mitosis. (**right**) Induction of apoptosis.

**Figure 5 ijms-24-06063-f005:**
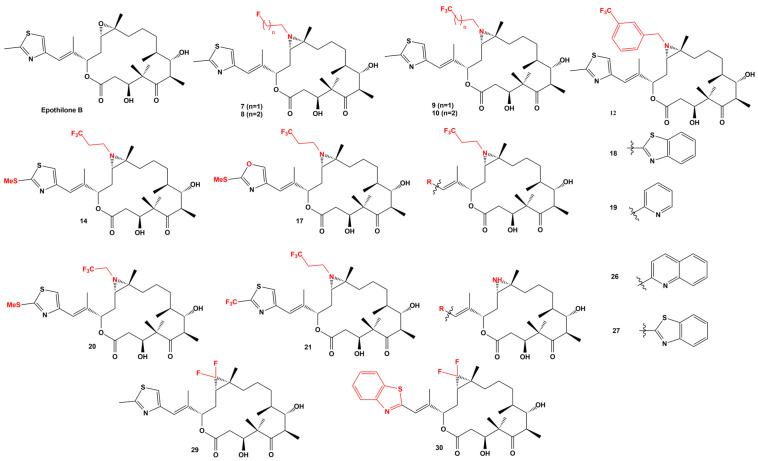
Chemical structures of the most active epothilone derivatives. These structures were published by [123]. Cytotoxic activities are reported in Table 4.

**Figure 6 ijms-24-06063-f006:**
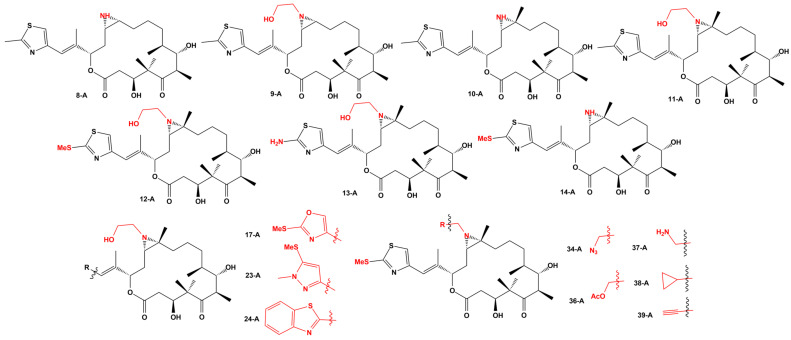
Structure of selected 12, 13-Aziridinyl epothilones with cytotoxic activity.

**Table 1 ijms-24-06063-t001:** Selected biotechnological methods in epothilone production.

Strains	Method	Results	Reference
*S. cellulosum*	Inactivation of the epoK gene by TALEN gene knockout system	Epothilone D yield increased to 34.9% and Epothilone B decreased to 34.2%	[33]
*Schlegella brevitalea* DSM 7029	Heterologous expression of different plasmids created by BioBricksTM and SSRTA methods	Enhancement of Epothilone B production to 82 mg/L in 6 days of fermentation	[34]
*S. cellulosum*	Optimization of parameters to 30 °C, initial pH = 7.4, speed of 200 r/min, inoculation of 10%, loading amount of 50/250 mL, fermentation 6 days, seed age of 60 h.	Increasing epothilone B production to 39.76 mg/L	[35]
*S. cellulosum*	Enhance the epothilone gene cluster with a novel promoter P3 by TALE-TF and CRISPR/dCas9	Epothilone B yield increased by 2.89- and 1.53-fold. Epothilone D yield improvement by 1.12- and 2.18-fold	[16]
*S. cellulosum*	Fermentation of *S. cellulosum* modified with plasmids pR6K-Amp-H.a-f-Ptet-H.a-r and pR6K-H.a-f-PBAD-H.a-r	Increasing the Epothilone B production to 93 mg/L	[36]
*Burkholderiales strain* DSM 7029	Electroporation of epothilone gene cluster 56 kb to DSM 7029, plus methylmalonyl-CoA and overexpression of tRNA genes	Increase the yields of epothilones production by 75-fold to 307 μg/L	[37]
*S. cellulosum*	Fermentation of immobilized *S. cellulosum* into porous ceramics	Increasing by 4-Folds the epothilone production to 90.2 mg/L	[38]

**Table 2 ijms-24-06063-t002:** Mechanism of action associated with apoptosis induced by different epothilones.

Epothilone	Cell Line	Mechanism	Reference
Epothilone A	Human neuroblastome CHP134 SK-N-SH C3H (ATCC 226 CCL) (wild type p53)	Apoptosis independent of p53.	[61]
Epothilones A and B	SKOV-3 human ovarian cancer.	Antiproliferative capacity of Epo A and Epo B six and five times higher than that of PTX, respectively Time-dependent induction of apoptosis and necrosis Cell death associated with decreased MMP, and ROS production	[55]
Epothilone B	NCI-H460 (H460) Human NSCLC cell lines	Late Activation of Caspases (cleavage of caspase-8)	[50]
Epothilone B	SKOV-3 human ovarian cancer.	Apoptosis induced mainly by the extrinsic pathway Increased cytosolic cytochrome c level after 4 h of treatment Increased intracellular calcium level > 20% after 24 and 48 h of exposure Increased TRAIL expression Activation of caspases-8 and -3 Cleavage of 116 kDa PARP to 25 kDa fragments	[56]
Epothilone B	OV-90 Human ovarian papillary serous adenocarcinoma.	Activation of receptors on the target cell surface Induction of apoptosis through a TRAIL- and caspase 8-dependent pathway (extrinsic pathway) Release of TNF-related apoptosis-inducing ligand (TRAIL) Immediate activation of initiator caspases 8 and 9, leading to the appearance of caspase 3 DNA fragmentation and reduced repair capacity	[58]
Iaxabepilone	Human epithelial ovarian tumor cell line 2008.C13 (cisplatinum-resistant variant)	Cytosolic accumulation of cytochrome C, Smac/DIABLO, and caspase-3-mediated PARP cleavage activity Increased DR4 and DR5 expressions Decreased intracellular levels of XIAP, cIAP and surviving Cytotoxic effects against cisplatin- and paclitaxel-resistant ovarian cancer cells	[60]
Patupilone (EpoB)	SK-N-SH IMR-32 Human neuroblastoma	Increased ROS generation; specifically, from mitochondria, after 2 h of treatment Accumulation of BIM in the mitochondrial compartment (2.4-fold) after only 6 h of treatment	[57]
Utidelone (UTD1)	RKO HCT116 CACO2 SW620 HCT15 (ABCB1 high-expression)	Mitochondrial pathway-dependent apoptosis > paclitaxel and 5-FU, especially in cells with high ABCB1 expression Increased caspase-3 activity and PARP cleavage Reduction of the mitochondrial membrane potential Release of mitochondrial cytochrome C Increased ROS production and activation of c-Jun N-terminal kinase (JNK) kinase Inhibition of tumor growth in a CRC xenograft mode	[54]

MMP: mitochondrial membrane potential, ROS: Reactive Oxygen Species.

**Table 3 ijms-24-06063-t003:** Cytotoxicity of ixabepilone against 21 tumor cell lines.

Cell Line	Ixabepilone IC50 (nM)	Cell Line	Ixabepilone IC50 (nM)	Cell Line	Ixabepilone IC50 (nM)
A2780/DDP-S	2.8	A2780/DDP-R	1.8	A2780/TAX-S	2.6
A2780/TAX-R	4.9	OVCAR-3	1.8	MCF-7	2.7
SKBR3	2.3	LNCAP	1.5	PC3	4.6
HCT116	2.6	HCT116/VM46	24.5	HCT116/VP35	2.0
LS174T	5.8	MIP	24.8	A549	5.2
LX-1	3.1	A431	1.4	CCRF-CEM	6.0
K562	2.9	M109	2.9	MLF	34.5

Median IC_50_: 2.9 nM.

**Table 4 ijms-24-06063-t004:** Cytotoxicity values for selected synthetic epothilone compounds, IC_50_ in nM.

Compound	MES SA DXE	MES SA DX	HEK 293T	SKBR3	SKOV3	HeLa
7	0.33	0.55	0.02	0.60	0.10	0.61
8	0.36	0.91	0.05	0.94	0.17	0.78
9	0.01	0.03	0.001	0.02	0.01	0.02
10	0.03	0.61	0.05	0.59	0.18	0.40
12	0.99	2.30	0.35			
14	0.48	0.51	0.05	0.85	0.16	0.52
17	0.35	0.63	0.05	1.03	0.12	0.86
18	0.43	0.46	0.05	0.72	0.08	0.43
19	0.44	0.66	0.06	1.35	0.30	0.80
20	0.52	0.44	0.04	1.57	0.26	1.26
21	0.28	0.65	0.05	0.49	0.13	0.28
26	0.92	20.62	0.52	2.01	2.02	2.46
27	0.78	3.01	0.30			
29	0.10	0.19	0.02	0.19	0.04	0.10
30	0.33	0.28	0.03	0.53	0.14	0.54
MMAE	0.46	113.7	0.10	0.10	0.09	1.17
EpoB	1.49	3.63	0.33	2.32	1.27	1.87
Ixabepilone	7.72	278.4	2.72	9.29	8.41	9.75

MMAE: monomethyl auristatin E.

**Table 5 ijms-24-06063-t005:** Cytotoxicity values for 12, 13-Aziridinyl epothilones in various cancer cell lines (IC_50_ in nM).

Compound	MCF-7	OVCAR-8	NCI/ADR-RES	MDA-MB-435	SNB-75	MES SA	MES SA DX	HEK 293
8-A	2.5	5.5	38	-	-	1.14	16.95	0.95
9-A	2.0	3.0	8.3	-	-	8.01	15.73	0.67
10-A	2.0	1.5	35	-	-	0.04	0.51	0.02
11-A	3.0	4.5	55	-	-	0.94	13.71	0.17
12-A	28	75	55	42	60	0.13	0.66	0.03
13-A	65	93	2800	20	130	0.078	0.85	0.058
14-A	4.0	16	8.8	4.5	11	0.28	5.66	017
17-A	7.5	25	6.5	3.5	13	0.02	1.11	0.05
23-A	78	10	7.5	12	10	0.02	37.76	0.24
24-A	11	23	630	3.5	12	0.18	1.32	0.06
34-A	13	15	3.2	7.3	22	0.24	0.52	0.10
36-A	5.5	18	7.0	3.5	23	0.29	0.86	0.07
37-A	14	63	70	15	23	0.108	11.98	0.079
38-A	18	15	18	9.5	16	0.056	1.257	0.051
39-A	30	18	7.0	3.5	31	0.23	0.45	0.09
Paclitaxel	7.8	26	4800	5.0	15	2.47	>400	1.76
MMAE	-	-	-	-	-	0.096	88.19	0.068
NAC	-	-	-	-	-	0.364	15.31	0.166

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
