# Peer review of "Epothilones as Natural Compounds for Novel Anticancer Drugs Development"

_ijms, 2023, doi:10.3390/ijms24076063_

Round 1

Reviewer 1 Report

This manuscript summarizes the developments of 16-membered macrolide, epothilones, against cancer with preclinical and clinical results. Manuscript is well-written and is recommended for publication after minor changes as suggested below.

Mention years for literature coverage in abstract

Add few more keywords

Maintain consistency in chemical drawings

Journal names in some references are abbreviated while in others given as full names; consider revision as per journal’s policy.

Author Response

Reviewer 1

This manuscript summarizes the developments of 16-membered macrolide, epothilones, against cancer with preclinical and clinical results. Manuscript is well-written and is recommended for publication after minor changes as suggested below.

  1. Mention years for literature coverage in abstract

Response: Required information has been added (Lines 17-18)

  1. Add few more keywords

Response: Three keywords were added: Epothilone derivatives, cytotoxicity and anticancer agents (Line 29).

  1. Maintain consistency in chemical drawings

Response: The color squares in figure 3 have been removed so that it looks similar to the rest of the chemical images.

  1. Journal names in some references are abbreviated while in others given as full names; consider revision as per journal’s policy.

Response: The bibliography was adjusted according to the reviewer's indications and the journal's standards

Reviewer 2 Report

I read with interest the manuscript by Villegas et al. Epothilones are a promising family of compounds with a similar mechanism of action as taxanes, but with improved solubility and less susceptibility to MDR mediated drug resistance.

The manuscript reads well and contains valuable information for the reader, however I found a major concern in the biochemical and structural part (section 2) that should be addressed before publication.

The binding mode of Epothilone to the taxane site both in dimers and in microtubules is well stablished by crystallography and cryo-EM. The structure proposed by Nettles et al (reference 20) does not correlate with the extensive characterization of the binding affinities (and cytotoxicities) of epothilones done by Nicolau in the early 2000 (https://www.sciencedirect.com/science/article/pii/S1074552104000250) and later by the Altmann´s group which are surprisingly not cited. The description of the interaction of epothilones with the taxane site should better be done on the basis of the crystallographic structure, which correlates well with the binding affinities, which are a far better parameter to quantify the interaction, binding affinities of a lot of Epothilone derivatives have been published by Altmann´s group.  I think that section 2 should be fully rewritten on the basis of the correct structure and the correlation of it with the binding affinities of derivatives for microtubules as published by Nicolau and Altmann´s groups.

Minor concerns.

From my personal experience neurotoxicity is a dose limitant toxicity in the treatment of patients with metastatic breast cancer. While patients treated with taxanes experience mild neuropathic symptoms, those treated with ixempra, show more frequently severe symptoms including incapacitant paraplegia. Although I might be biased by my personal experience, I am sure that many other colleagues are concerned by the very same problem which can be attributed to the ability of epothilones to cross the BBB. I find that it could be of interest for the reader to add the review a section describing the recent findings of the neurotoxic effects of epothilones, (including clinical data), which for my surprise are described to be mild.

Author Response

We appreciate the comments done by the reviewer;  here we change and explain better way the interaction of epothilones with tubulin in a new paragraph.

A chapter was added with the requested information on the toxicity profile of the epothilones with emphasis on Ixabepilone, the most clinically used epothilone

Reviewer 3 Report

1. English writing should be checked.
2. The term "scaffolds" is not appropriate for this article because scaffolds are mainly used in tissue engineering. The compound can be used instead of the scaffold.
3. In line 52, a small explanation of the nature and source of paclitaxel should be given. There are some useful resources about paclitaxel that are recommended to visit, for example:
https://doi.org/10.1016/j.ijpharm.2017.05.016
https://doi.org/10.1080/10408347.2017.1416283
https://doi.org/10.1016/S1875-5364(20)60032-2
10.17305/https://doi.org/bjbms.2016.674

4. The search strategy is not mentioned.
5. More references should be given in line 196. For example PMID: 27114800
  https://doi.org/10.1002/cmdc.200600308
6. Reference should be given in line 198.

7. At the end of the introduction, write the aim of writing this article.

8. Section "Epothilone synthesis" can be moved to the next section after the introduction.

Author Response

The authors sincerely thank the reviewers for their valuable comments on our work. In the revised version of the manuscript, we carefully addressed each comment and provided a detailed response in this document. Additionally, careful English language and grammar revisions were conducted to correct the minor mistakes pointed out by the referees after the revision process.

Round 2

Reviewer 2 Report

The authors have adequately addressed my concerns. The manuscript can be published as it is.